

# Investigating passive eDNA samplers and submergence times for marine surveillance

Ulla von Ammon[1], Gert-Jan Jeunen[2], Olivier Laroche[1],
Xavier Pochon[1,3], Neil J. Gemmell[4], Jo-Ann L. Stanton[4] and
Anastasija Zaiko[5]

[1] Marine Biosecurity, Cawthron Institute, Nelson, Tasman, New Zealand
[2] Department of Marine Science, University of Otago, Dunedin, Otago, New Zealand
[3] Institute of Marine Science, University of Auckland, Auckland, Tasman, New Zealand
[4] Department of Anatomy, School of Biomedical Sciences, University of Otago, Dunedin, Otago, New Zealand
[5] Sequench Ltd, Nelson, Tasman, New Zealand

Corresponding author
Ulla von Ammon,
ulla.vonammon@cawthron.org.nz

## ABSTRACT

Passive environmental DNA (eDNA) samplers offer a cost-effective and scalable approach to marine biodiversity monitoring, potentially aiding detections of non-indigenous species. This study explored the efficiency of passive eDNA samplers to detect a variety of globally problematic marine invasive species in field conditions: *Sabella spallanzanii*, *Styela clava*, *Bugula neritina* and *Undaria pinnatifida*. Four passive sampler substrates, nylon filters, positively charged nylon discs, nylon mesh, and artificial sponges, were tested across six submergence times, ranging from 10 to 720 min, against standard filtration-based approaches. Our results demonstrated that passive samplers could achieve comparable or even higher eDNA yields than traditional active filtration methods, indicating their potential for biosecurity surveillance. Species-specific droplet-digital PCR (ddPCR) assays provided sensitive and quantifiable eDNA signals, though assay validation remains crucial to avoid false negatives. Significant variation in eDNA signal detection highlighted the importance of considering both material selection and submersion time, depending on the targeted organisms. Furthermore, 18S rRNA metabarcoding was undertaken to assess how the overall detected biodiversity might interfere with species-specific detections. Certain sessile organisms, such as ascidians and polychaetes, dominated early representation on the passive filters but did not interfere with species-specific detection. By optimizing material selection, submersion time, and assay validation, passive eDNA sampling can enhance the sensitivity and reliability of eDNA-based monitoring, contributing to improved marine biosecurity and conservation efforts.

# INTRODUCTION

Environmental DNA (eDNA) technologies have emerged as a powerful biomonitoring tool for aquatic environments (*Cristescu & Hebert, 2018*; *Ficetola et al., 2008*; *Takahashi et al., 2023*). For example, fish-specific genetic marker regions are being used to undertake biodiversity baseline surveys in lakes, rivers and the ocean (*Gold et al., 2021*), bacterial

community shifts are being studied through eDNA to assess the impact of anthropogenic activities on benthic ecosystems (*Nigel, Wood & Pochon, 2018*), and seawater eDNA data are being screened for the early detection of invasive species in coastal biosecurity surveillance (*von Ammon et al., 2023*; *Darling et al., 2020*). For all these applications, eDNA offers a non-invasive, cost-effective and sensitive approach for depicting biodiversity across the tree of life by analysing the genetic material organisms shed into the water in the form of faeces, saliva, tissue cells, gametes or extracellular DNA (*Taberlet et al., 2012*).

The conventional workflow for aquatic eDNA biomonitoring begins with collecting between 15 and 10,000 mL of water per sample, followed by eDNA concentration through manual or vacuum filtration (*Pilliod et al., 2014*; *Rees et al., 2014*). More novel technologies filter large amounts (up to 5 m$^3$) of seawater by: towing plankton nets in the open sea (*von Ammon et al., 2020*), adapting underway seawater systems (*Jeunen et al., 2024*), or harnessing the power of naturally occurring marine filter feeders (*Jeunen et al., 2023*). While effective, many of these approaches are time consuming, require specific equipment or are prone to cross-contamination due to the involvement of several handling steps (*Buxton et al., 2021*). These constraints limit upscaling of eDNA surveillance efforts or restrict meaningful sample replication (*Scriver et al., 2024a*; *Jeunen et al., 2022*). However, fine-scale spatio-temporal surveys across large areas are a prerequisite for successful monitoring and essential to obtain the necessary data for informed ecosystem management (*von Ammon et al., 2023*).

Among the advancements in this relatively young field of research (*Pochon et al., 2023*; *Connelly et al., 2021*), passive eDNA samplers have garnered considerable attention as a simple, cost and time-efficient approach (*Bessey et al., 2021*; *Kirtane, Atkinson & Sassoubre, 2020*). Rather than actively filtering water, passive samplers use a variety of adsorbent materials, including simple cellulose nitrate filters (*Chen et al., 2022*) to more complex Montmorillonite clay matrices (*Kirtane, Atkinson & Sassoubre, 2020*), to accumulate eDNA from the surrounding aquatic environment (*van der Heyde et al., 2023*). Mesocosm and in-field studies have shown promising results in terms of their efficiency and accuracy to detect communities and target organisms compared to the current gold standard of active filtration (*Jeunen et al., 2022*; *Chen et al., 2022*). Additionally, *Bessey et al. (2022)* showed that after just 5 min of deployment time, materials like sponge and cellulose retrieved up to 100% of the fish biodiversity identified *via* conventional tools. Therefore, passive eDNA samplers could enable upscaled sampling through increased spatio-temporal replication, without the need for complex and laborious sampling protocols. Furthermore, it can be performed without personnel being present throughout the sampling process, offering a more hands-off, cost-effective, and user-friendly workflow which also significantly reduces contamination issues (*Chen et al., 2022*).

Despite their high potential, the efficiency of passive samplers can be substantially affected by the duration of deployment (*Chen et al., 2022*). To strike a balance between submerging the passive samplers long enough for eDNA accumulation to occur while limiting eDNA degradation, the effect of submergence times on eDNA detection needs to be investigated (*Chen et al., 2024*).

In this study, we aimed to explore the efficiency of passive eDNA samplers, with a focus on the overall community sampled and the detection of non-indigenous species. We tested the optimal submergence time of four passive sampler substrates based on the study of *Jeunen et al. (2022)*, including nylon filters (5 μm pore size), positively charged nylon discs (0.45 μm pore size), nylon mesh (20 μm pore size), and artificial sponges. We tested six submergence times for each substrate, from 10 to 720 min (12 h), assessing eDNA-based detection of four locally abundant invasive species (see Marine Biosecurity Porthole, https://marinebiosecurity.org.nz/) using species-specific droplet-digital PCR assays of the Mediterranean fanworm *Sabella spallanzanii* (Gmelin, 1791), the Japanese kelp *Undaria pinnatifida* (*Perez, Lee & Juge, 1981*), the clubbed tunicate *Styela clava* (Herdman, 1881) and the bryozoan *Bugula neritina* (Linnaeus, 1758). Additionally, we performed 18S rRNA metabarcoding on all samples to assess potential selective capture of different taxa on the tested samplers, and how the overall detected biodiversity might interfere with species-specific detection.

By determining the best combination of passive sampler substrates and their submergence time for both species-specific detection and community characterization, we seek to enhance the effectiveness of eDNA passive sampling for marine biomonitoring.

# MATERIALS AND METHODS

## Field deployment

A total of 72 samples (excluding negative controls) were deployed from pontoon B65 (−35.836802, 174.468679) in Marsden Cove, Whangārei, Aotearoa New Zealand from the 19th–20th of June 2021 for a 12 h period. Cawthron holds a Special Permit with the Ministry for Primary Industries (SP822-2) that allows the taking of fish, aquatic life and seaweed for the purposes of education and investigative research. Four substrates were explored; nylon filters (5 μm pore size; Sterlitech, Auburn, WA, USA), positively charged nylon discs (0.45 μm pore size; Biodyne™; Thermo Fisher Scientific, Waltham, MA, USA), UV-treated nylon mesh cut in 50 × 50 mm pieces (20 μm pore size; NITEX™; Sefar Ltd., Thal, Switzerland), sterile Whirl-Pak® Speci-Sponges® (Nasco, Merck, Germany), and additionally the sponge water was squeezed from the Whirl-Pak® Speci-Sponges® and DNA extracted directly from the supernatant as in *von Ammon et al. (2018a)* (Fig. 1). Substrates were attached with cable ties to a bottle carrier and deployed at ~1 m below the water surface. Three replicates of each substrate were retrieved at 10, 30 min, 1, 3, 6 and 12 h after deployment. Upon retrieval, the membranes were cut in half and placed in 2 ml Eppendorf tubes prefilled with LifeGuard Soil Preservation Solution (Qiagen, Hilden, Germany), except Whirl-Pak® Speci-Sponges® which were placed into individual bags due to their large size and put on ice.

The water isolated from the Whirl-Pak® Speci-Sponges® was later analyzed as an additional individual sample type.

For reference (waterborne eDNA detection), triplicate 500 ml seawater samples were collected directly in front of the bottle carrier at the start (0 h) and end (12 h) of passive eDNA sampler deployment. These samples were vacuum filtered through sterile Whatman filters (pore size 0.45 μm; Merck KGaA, Darmstadt, Germany) placed in a

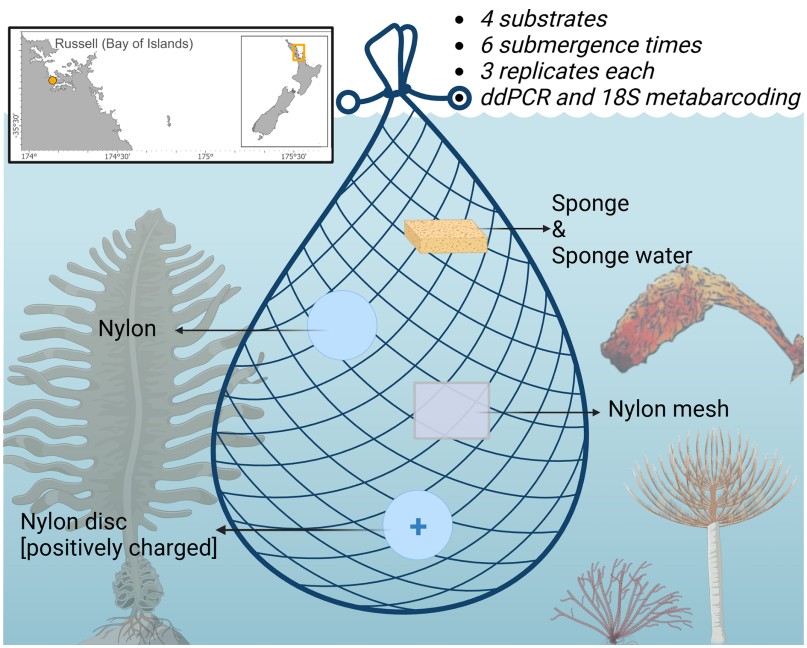

**Figure 1 Schematic of the experimental design highlighting the four sampling substrates nylon filters (5 μm pore size), positively charged nylon discs (0.45 μm pore size), nylon mesh (20 μm pore size), and artificial sponges.** Displayed are the non-indigenous species (NIS) surveyed by droplet digital PCR in this study. The Japanese kelp *Undaria pinnatifida* (bottom left), the brown bryozoan *Bugula neritina* (bottom middle), the Mediterranean fanworm *Sabella spallanzanii* (bottom right) and the clubbed tunicate *Styela clava*. (right side). Figure adapted from *Jeunen et al. (2022)* and created with BioRender.com. 

Sterifil filtration system (Merck, Darmstadt, Germany) using a 12V Seaflo 21 Series Water Pressure Pump (SEAFLO, Fujian, China). Each filter was stored in a 2 ml Eppendorf tube containing 1.5 ml of LifeGuard Soil Preservation Solution (Qiagen, Hilden, Germany).

Negative field controls (blanks) consisted of (i) 500 ml ultrapure water filtered through the same 0.45 μm filter type as water field controls, and (ii) all materials used for passive filtration were taken out briefly in the field but then immediately placed into clean tubes or bags.

## Laboratory processing

All samples were extracted using a modified protocol of the Qiagen DNeasy Blood & Tissue Kit (*Jeunen et al., 2019*). Materials were placed into new 2 ml Eppendorf tubes with 0.5 mm Zirconia/Silica Beads (Dnature, Aotearoa New Zealand) and 720 μL ATL buffer (Qiagen, Hilden, Germany), and 80 μL Proteinase K (Qiagen, Hilden, Germany) added. The original 2 ml Eppendorf tubes with 1.5 ml of LifeGuard Soil Preservation Solution (Qiagen, Hilden, Germany) were centrifuged at 10,000 g for 5 min, supernatant discarded, and the remaining pellet re-eluted with ATL buffer and transferred to the bead beating tube. Samples were homogenized at maximum speed (1,500 strokes/min) *via* bead beating for 2 min (1600 MiniG Spex SamplePrep) and incubated overnight at 56 °C. The following day samples were vortexed for 1 min, followed by a centrifugation step for 1 min at 6,000 g.

600 μL of supernatant was transferred to a new 2 ml Eppendorf tube, after which the standard Qiagen DNeasy Blood & Tissue Kit protocol was followed.

## Species-specific assays

Droplet digital polymerase chain reactions (ddPCRs) were run on a QX200 Droplet Digital PCR System™ (Bio-Rad, Hercules, CA, USA) for four targeted assays amplifying the Mediterranean fanworm *Sabella spallanzanii*, the clubbed tunicate *Styela clava*, the Japanese kelp *Undaria pinnatifida* and the brown bryozoan *Bugula neritina*. *Sabella spallanzanii* specific Cytochrome c oxidase subunit 1 (COI) primers forward Sab3F: 5-GCT CTT ATT AGG CTC TGT GTT TG-3 and reverse Sab3R: 5′-CCT CTA TGT CCA ACT CCT CTT G-3′ and Taqman probe Sab3 5′-FAM/AAA TAGT TCA TCC CGT CCC TGC CC/BkFQ-3′ were used as in *Wood et al. (2017)*. For *Styela clava*, we used the COI primers forward SC1F 5′-TCCGGCGGTAGTCCTTTTATT-3′ and reverse SC1R 5′-GAGATCCCCGCCAAATGTAA-3′ as well as and the TaqMan® probe SC1 5′-TTA GCTAGGAACTTGGCCCA-3′, as in *Gillum (2014)*. For both assays, each reaction included 450 nM (1 μl) of each primer and probe, 1 × ddPCR Supermix for probes (10 μl, no UTP; Bio-Rad, Hercules, CA, USA) and 3 μl DNA and 5 μl sterile water for a total reaction volume of 21 μl. Amplification followed the cycling protocol: 95 °C for 10 min, 40 cycles of 94 °C for 30 s, 60 °C for 1 min, and 98 °C for 10 min. For *Undaria pinnatifida*, we used the COI forward primer U_pinnatifida2F 5′-TACAGCAATGTCTGTTTTTATCC-3′, reverse primer U_pinnatifida2R 5′-ACATTATACAACTGATGATTTCCC-3′ and probe U_pinnatifida2Probe_BHQ2 5′-FAM-ATTGCAATTAGCTAGCCCTG/ 3BHQ_2-3′ (*Bott, Giblot-Ducray & Deveney, 2015*). Each reaction included 450 nM of each primer and probe (1 μl), 1× ddPCR Supermix for probes (10 μl, no UTP; Bio-Rad, Hercules, CA, USA) and 3 μl of DNA and 5 μl sterile water for a total reaction volume of 21 μl. The amplification cycling protocol was as follow: 95 °C for 10 min, 40 cycles of 95 °C for 30 s, 57 °C for 30 s, and 98 °C for 10 min. For *Bugula neritina*, an Evagreen assay was used with the COI primers BuNe_SF 5′-GGTACATTATACTTTTTATTTGGAC-3′ and BuNe_SR 5′-CCCCCA ATTATAACTGGTATG-3′ primers as in *Kim et al. (2018)*, and conditions adapted as in *Scriver et al. (2024a)* to reactions that included 450 nM of each primer, 1× ddPCR Evagreen Supermix (no UTP; Bio-Rad, Hercules, CA, USA) and 3 μl of DNA and 6 μl sterile water for a total reaction volume of 21 μl. The amplification protocol was as follow: 95 °C for 5 min, 40 cycles of 95 °C for 30 s, 57 °C for 1 min (2′ ramp), and 90 °C for 5 min. Each well of the plate was then individually analyzed on the QX200 instrument to establish the threshold value separating negative and positive droplets and perform absolute quantification of target DNA (see *Chiu et al., 2017*). A negative control (DNA-free water) and positive control (consisting of extracted DNA from tissue of the targeted species) were included on each plate.

## 18S rRNA metabarcoding

For metabarcoding analyses, polymerase chain reactions (PCRs) were performed to amplify the V4 region of the nuclear small ribosomal subunit (18S rRNA) gene. The primers for the 18S rRNA gene were Uni18SF: 5′-AGG GCA AKY CTG GTG CCA GC-3′

and Uni18SR: 5′-GRC GGT ATC TRA TCG YCT T-3′ (*Zhan et al., 2013*) with Illumina overhang adaptors (*Kozich et al., 2013*). Amplifications were undertaken on an Eppendorf Mastercycler (Eppendorf, Hamburg, Germany) in a total volume of 50 µl using 25 µl of MyFi Mix (Bioline, Meridian Bioscience, Cincinnati, OH, USA), 1 µl of each primer, 20 µl of DNA-free water, and 3 µl of template DNA. Thermocycling conditions were as follows: 95 °C for 2 min, followed by 40 cycles of 95 °C for 20 s, 52 °C for 20 s, 72 °C for 20 s, and a final extension of 72 °C for 10 min. Negative PCR controls (3 µL of DNA-free water) were run alongside the samples in the PCR. Amplicons were cleaned and normalized using the SequalPrep Normalization kit (Thermo Fisher Scientific, Waltham, MA, USA) resulting in a concentration of ~1 ng/µl. Amplicons were sent to Sequench Limited (Nelson, Aotearoa New Zealand) for indexing with Nextera XT (Illumina, San Diego, CA, USA) kit and high-throughput sequencing on an Illumina MiSeq platform. The library pool was diluted to a final loading concentration of 6 pM with a 15% PhiX spike and sequenced using MiSeq Reagent Kit v3-600 cycle (2 × 301 bp) (Illumina, San Diego, CA, USA). A sequencing control (DNA-free water) was added to each sequencing run.

## Bioinformatics

FASTQ files were demultiplexed and primers removed using Cutadapt (version 3.7; *Martin, 2011*), using a minimum overlap of 15 bp. Forward and reverse sequences were truncated on their 3′ end at 225 and 216 bp for forward and reverse reads, respectively, to remove low quality sections of the reads. Sequences were quality filtered and denoised using the default parameters of the 'DADA2' R package (version 1.21; *Callahan et al., 2016*). Forward and reverse reads were merged using a minimum overlap of 10 bp, and chimeric sequences removed using the consensus method of 'DADA2'. Remaining sequences were taxonomically assigned with the SILVA database (version 132) using a Naive-Bayesian classifier and with blastn and megablast against the GenBank nt database using default settings of blastn_taxo_assignment from the biohelper R package (https://github.com/olar785/biohelper, following *Laroche et al. (2020)*. Taxonomic assignments for the three approaches were merged using the taxo_merge function of biohelper, where taxonomy is first normalized using NCBI's curated taxonomy database, using the highest taxonomic assignment (resolution) across methods if there is consensus among them (>50%), across assigned taxonomic ranks. In case of discrepancy, the last common ancestor is assigned. Sequences found in blanks were investigated and potential contamination removed using the MicroDecon R package (version 1.0.2, *McKnight et al., 2019*). For removal of potential contamination, microDecon compares the taxa (ASVs) present in the negative controls to those in the study samples. Those present in both the negative controls and the study samples are flagged as potential contaminants. The microDecon algorithm assumes that the proportion of each putative contaminant in the study samples can be approximated based on its relative abundance in the negative controls. For each shared ASVs (present in both controls and samples), it calculates the expected contaminant proportion in the sample which is then subtracted. The advantage of this method over simply removing all ASVs found in blanks is that it enables keeping relevant and often abundant/important taxa that may have been detected in blanks. In this
study, we have used the microDecon's default parameters and have included ASVs identified in negative controls in the Supplemental Material (see Table S2).

Raw sequence reads were deposited in the NCBI short read archive under the accession number PRJNA1154297.

## Statistical analysis

Bar plots were used to visualize species-specific ddPCR detections (copies/μl) using ggplot2 package (v3.5.0.; *Wickham, 2016*). The mean value of active filtration was overlayed as a horizontal line including standard deviation. The relationship between deployment time and detected DNA (copies/μl) *via* ddPCR for the four investigated non-indigenous species (NIS) was tested per species per matrix using a generalized linear model (GLM) with an exponential link function and visualized using the ggpubr R package (version 0.6.0; *Kassambara, 2018*).

For the 18S rRNA metabarcoding dataset, sequences identified as non-eukaryotic or non-marine taxa were discarded, and rarefaction curves performed on the remainder of the data to investigate sequencing depth per sample using the 'rarefy_even_depth' function (Fig. S1, phyloseq v.1.40–1 R package). Samples with read sum <1,000 reads were removed. The effect of deployment time and matrix on overall amplicon sequence variants (ASV) richness and those solely associated to NIS was investigated and visualized in scatter plots including $R^2$ and *p*-values with ggpubr. Eukaryotic community composition was visualized with bar plots using the biohelper R package (https://github.com/olar785/biohelper), following *Laroche et al. (2020)*, for overall biodiversity at phylum and family level. For subsampled NIS taxa at genus and species level, the non-rarefied fasta reads for 18S rRNA were screened with the Pest Alert Tool (https://gitlab.com/cawthron-public/marine-biosecurity-toolbox/pest-alert-tool, *Zaiko et al., 2023*) using Minimum % sequence identity match = 99.5% and Minimum sequence length (300 bp). Beta-diversity was initially assessed with a principal component analysis (PCA) on the 18S rRNA dataset with deployment time displayed with an ordisurf layer. In addition, the correlation of deployment time with biological assemblages is showed with an arrow, with Pearson (r) and *p*-value displayed. The effect of deployment time and matrix was tested with a permutational analysis of variance using the vegan R package (version 2.6.9; *Oksanen et al., 2018*). The progression of beta-diversity dissimilarities for each matrix and along submersion time was plotted in a line diagram using phyloseq_group_dissimilarity function of the metagMisc R package (v 0.5.0).

## RESULTS

### Species-specific detections of key non-indigenous species

From the four species-specific ddPCR assays, only *Undaria pinnatifida* was not detected, neither with any of the passive samplers nor with the vacuum filtering, the later undertaken for reference purposes. However, *Sabella spallanzanii*, *Styela clava* and *Bugula neritina* were detected throughout submersion times from the different substrates but displayed variable detection signals and patterns (Figs. 2 and 3). *Sabella spallanzanii* showed the strongest amplification signals of the tested assays, ranging from a maximum

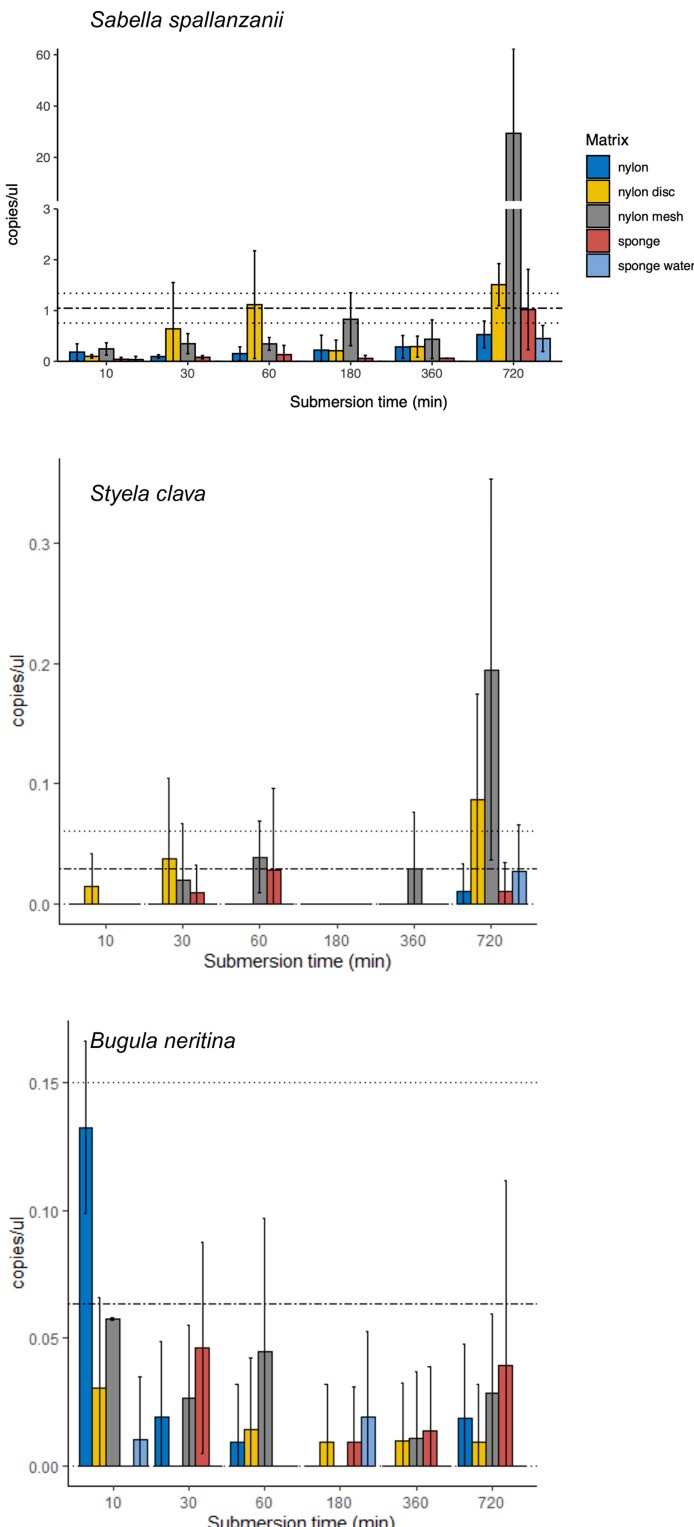

**Figure 2 Bar plots of copy numbers/µl per species for the different passive samplers (nylon, nylon disc, nylon mesh, sponge and sponge water) for 10, 30, 60, 180, 360 and 720 min after deployment.** Standard deviation per matrix and deployment time is indicated as per vertical line. Mean value with standard deviation for the filtered seawater samples is added as horizontal dashed lines.

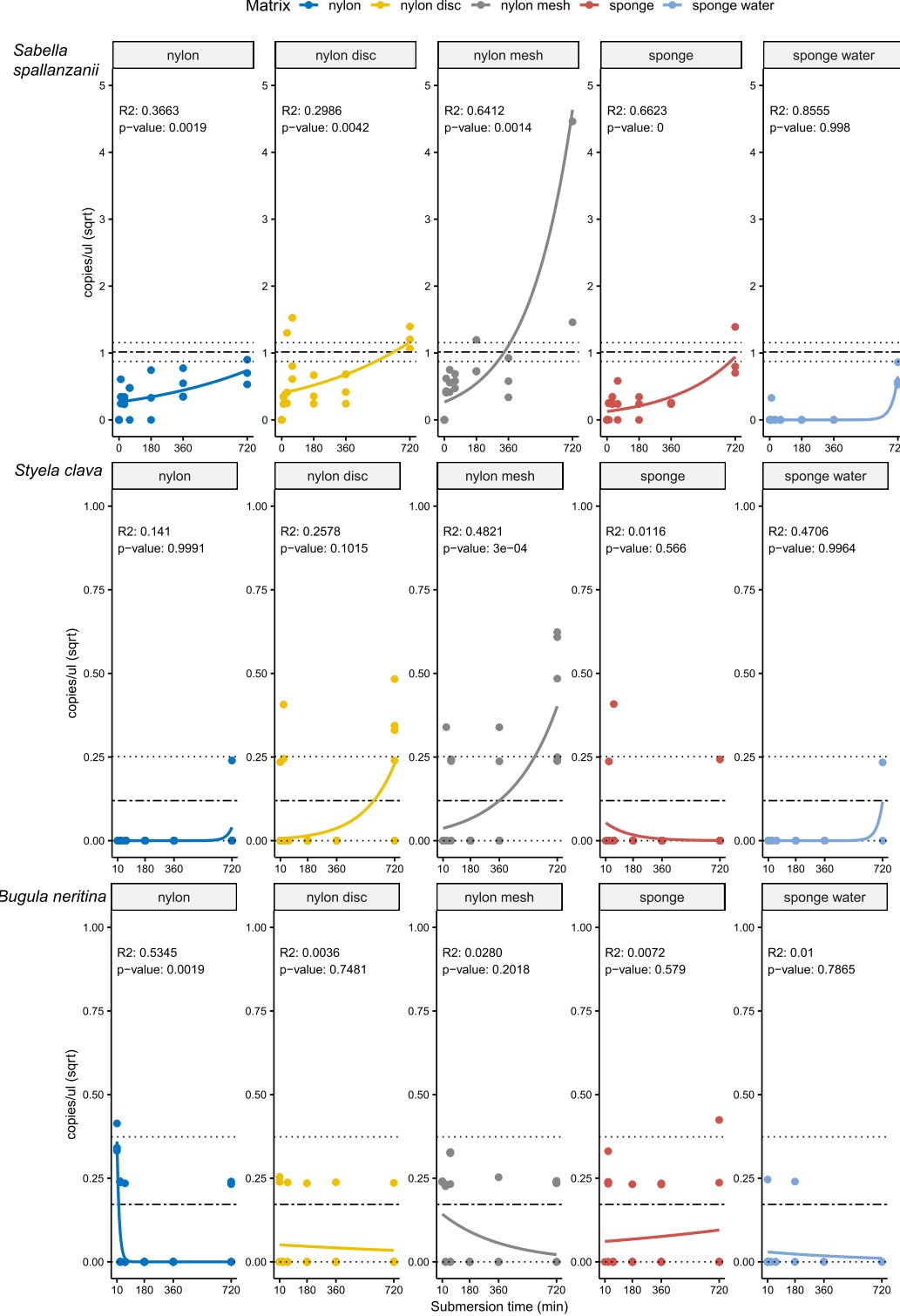

**Figure 3 Generalised linear models with exponential regressions for the species-specific assays** ***Sabella spallanzanii, Styela clava*** **and** ***Bugula neritina*** **in copies/µl** ***vs*** **sampling time points at 10, 30, 60, 180, 360 and 720 min.** Plots are divided into different sampling materials (coloured) and $R^2$ and $p$-values given for each plot. Mean value with standard deviation for the filtered seawater samples were added as horizontal dashed and dotted lines, respectively.

value of 66 copies/µl to a median value of 0.1185 copies/µl. Interestingly, vacuum filtration for *S. spallanzanii*, which was only undertaken at time point 0 and 720 min (12 h), showed consistent copy numbers (mean = 0.9725 ± 0.29 copies/µl). All passive samplers but not sponge water showed a significant positive temporal trend of copy numbers/µl with time ($p \leq 0.05$, Fig. 3), with nylon discs reaching the reference mean value of vacuum filtered samples after 60 min (Fig. 2). After 720 min, nylon disc exceeded the reference value with 2.33 copies/µl and nylon mesh with a maximum value of 66.1 copies/µl (median 0.3 ± 14.74 copies/µl).

For *Styela clava*, species-specific detection was more sporadic across sampling events with 0 detections for all samplers at 180 min (Fig. 2). This species was detected only in 50% of the water samples, with relatively low signal (mean = 0.027 ± 0.03 copies/µl). The passive sampler nylon mesh still displayed a significant increase in copy numbers over time ($p \leq 3e{-}04$, Fig. 3). Nylon disc and nylon mesh exceeded the reference values from active filtration after 30 min and 1 h, respectively. At the final sampling time point (720 min), all passive sampling matrices showed detections (Fig. 2), with nylon disc (0.08 copies/µl), nylon mesh (0.19 copies/µl) and sponge water (0.02 copies/µl) reaching or exceeding the reference waterborne values.

The *B. neritina* assay yielded positive detections across the different sample types and time points (Fig. 2) but remaining at the lower end of the overall copy numbers (0.05–0.2 copies/µl) and showing no significant pattern of eDNA signal accumulation over time except a decreasing trend for nylon, highly influenced by the high initial copy numbers ($p \leq 0.0019$, Fig. 3). The reference value from water samples (mean = 0.06 ± 0.08) was exceeded with nylon, nylon disc, nylon mesh and sponge within 10 to 60 min. All passive samplers except for sponge water showed detection signals after 720 min (Fig. 2).

## Metabarcoding data

High-throughput sequencing yielded a total of 12,318,746 18S rRNA reads, of which 85% remained after primer trimming, 44.1% after denoising, 40% after merging, and 36.1% after chimera removal (Table S1). Removing potential contamination from negative controls (see Table S2) and unidentified amplicon sequence variants (ASVs) resulted in a total of 4,076,470 sequence reads (mean of 42,463 reads per sample) and a total of 4,046 ASVs. Rarefaction curves plateaued before 10,000 reads, therefore samples were rarefied to the minimum number of reads per sample (19,470 reads) prior to diversity analysis (see Fig. S1).

The overall ASV richness displayed significant decreasing temporal trends for all passive samplers, including sponge water ($p \leq 0.05$, Fig. 4A). Similarly, while not significant, a negative trend was observed for the NIS ASVs except for the sponge water samples, which indicated some increase over time, also not significant (Fig. 4B).

Looking at the taxonomic composition across sampling times and sample types, a clear difference could be observed between filtered water samples and the passive sampling (Fig. 5A). The majority of taxa sampled by water filtering was assigned to the copepods Calanoida, Oithonidae and Styelidae (Chordates), with a clear shift to a different family of copepods, Tachidiidae, observed at 720 min compared to the beginning of the experiment.

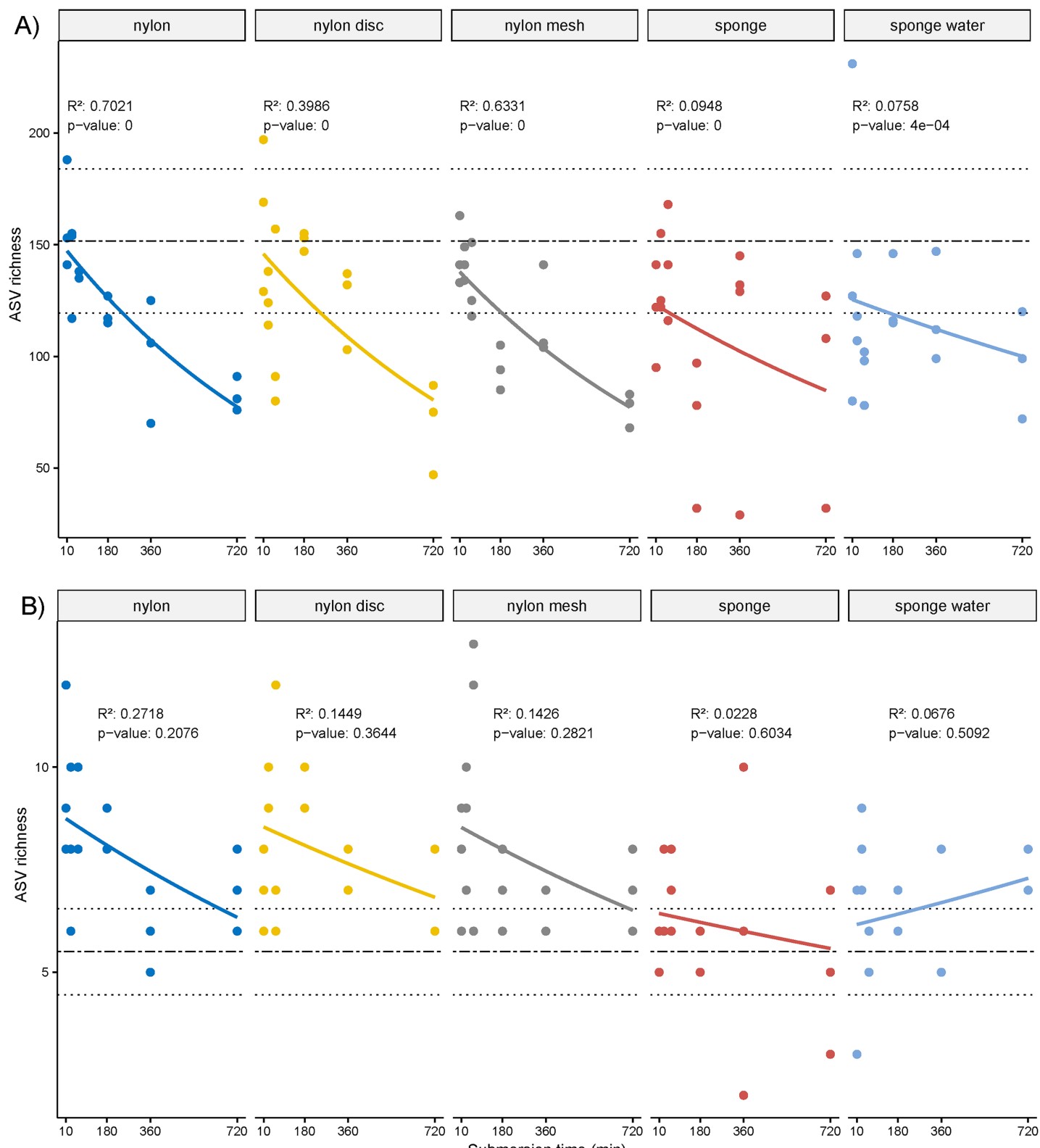

**Figure 4 Scatter plot of amplicon sequence variant (ASV) richness and for non-indigenous taxa.** (A) Scatter plot of amplicon sequence variant (ASV) richness across deployment time with exponential regression and associated statistics (R² and p-value). (B) Based on ASVs identified as non-indigenous taxa.

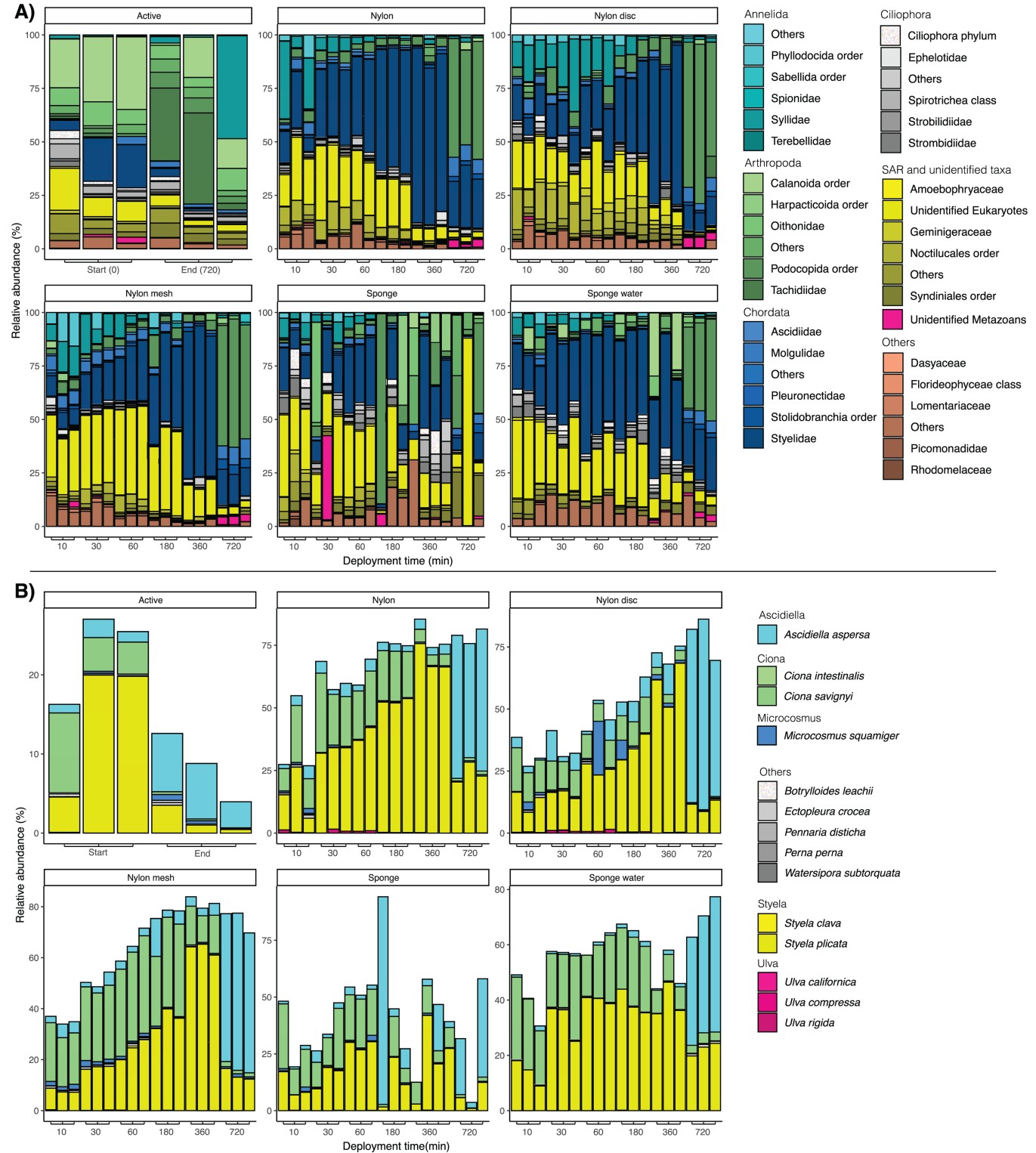

**Figure 5** (A) Bar plots of taxonomic composition at phylum and family level per matrix and deployment time. (B) Bar plots of taxonomic composition at species level, focusing solely on taxa identified as non-indigenous species (NIS). Relative abundance was calculated at species level and the data subsequently filtered to only keep sequences identified as NIS. NIS were retrieved from https://gitlab.com/cawthron-public/marine-biosecurity-toolbox/pest-alert-tool. Note: y-axis values vary between facets.

For the passive sampling, a successive shift through time with a clear dominance in relative abundance of Styelidae and unidentified Eukaryotes was reported (Fig. 5A). In addition, the relative abundance of the polychaete family Syllidae substantially increased from 10 to 30 min, before decreasing at later sampling time points. After 720 min, Podocopida (Ostracoda) became the most abundant taxonomic group, particularly for all nylon samplers.

Looking at the taxonomic composition of exclusively non-indigenous species, filtration and passive sampling approaches appeared very similar (Fig. 5B). Earlier samples (10 min–6 h) were dominated by *Ciona savignyi* and *Styela plicata*. After 720 min, *Ascidiella aspersa* clearly dominated across all samples. In filtered seawater samples, relative abundance of NIS decreased between the first and last sampling point, while for nylon, nylon disc, nylon mesh and sponge water, relative NIS abundance increased over time. The relative abundance values varied more drastically and appeared less consistent among sampling time points for the sponge sampler compared to the other passive samplers (Fig. 5B).

A permutational analysis of variance (PERMANOVA) performed to assess the effect of submersion time and sampling method among passive samplers confirmed significant effects of both factors and their interaction ($p \leq 0.001$, Table 1). When tested separately, the effect of submersion time remained significant for all sample types ($p \leq 0.001$, Table S3, Fig. S2).

In Fig. 6, the progression of beta-diversity dissimilarity shows an increasing trend over time for all matrices (10–720 min), although for nylon it nearly reaches a plateau at 360 min. While all nylon matrices very consistently increase in beta diversity dissimilarities, the sponge matrix shows a drop of dissimilarity at 60 and 360 min, while the sponge water steeply increases in dissimilarity until 60 min, drops slightly at 180 min and then increases again until 720 min. The beta diversity dissimilarity for the nylon matrix increased gradually from 30 to 360 min, with a more significant increase observed between 360 and 720 min. Overall, dissimilarity in community composition at the end of the experiment for passive samples was substantially higher than that of the active filtration samples (mean = 0.63, sd = 0.04; Table S4).

## DISCUSSION

In this study, our primary objective was to explore the efficiency of passive eDNA samplers across different deployment times, with a focus on detecting marine non-indigenous species (NIS), while also investigating changes in the overall eukaryote (18S rRNA) community over time.

The selection of the experimental substrates was based on results from *Jeunen et al. (2022)*, where different nylon matrices delivered among the strongest signals in fish diversity and richness. However, delicate nylon materials can be difficult to secure to a holding structure in open water, sometimes requiring protective enclosures to mitigate sample loss (*Chen et al., 2024*; *Maiello et al., 2022*). Such protective measures were not used in this study. The nylon membranes were freely deployed as a barrier may inadvertently

**Table 1 Permutational analysis of variance (PERMANOVA) on the eukaryotic community (18S rRNA).** Significant *p*-values are in bold.

| Term | df | $R^2$ | *p* value |
|---|---|---|---|
| Time | 1 | 0.20 | **0.001** |
| Matrix | 5 | 0.20 | **0.001** |
| Time * matrix | 5 | 0.07 | **0.001** |
| Residual | 84 | 0.53 | |

hinder molecular particle capture and affect eDNA adsorption rates (*Verdier et al., 2022*). In *Jeunen et al. (2022)* and *Jeunen et al. (2024)*, artificial sponges also delivered high detection signals and generally represented a more robust porous matrix, which could be easily deployed in high-flow water environments such as a marina, where our experiment was conducted (*Bessey et al., 2022*). Commonly used cellulose and glass fibre filters were not considered here because they tend to perform poorly and degrade more rapidly in the marine environment (*Jeunen et al., 2022*).

While the literature compares eDNA signal detection for multiple different matrices and their associated biodiversity, investigations of submersion times for passive samplers remain either untested or yield inconsistent results (*Bessey et al., 2022*; *Chen et al., 2024*; *Chen et al., 2022*), leaving the issue of what is optimal open. The optimal deployment time for passive samplers is likely influenced by various factors, including physico-chemical properties of the materials, which affect how they attract and retain eDNA molecules (*Zhang et al., 2024*), and their efficiency in releasing DNA during extraction processes (*Jeunen et al., 2022*). This study demonstrated that the fine nylon filters seemed to provide the best effective balance between retaining eDNA material over time, while being easy to process in the laboratory (see Figs. 2–4).

Other factors can substantially affect signal detection such as environmental conditions and the target organisms' shedding rate, itself influenced by life stages, size, skin/scale properties, surface area to biomass ratio, and their behaviour such as stress, mobility, feeding activities, and metabolic rates (*Scriver et al., 2024b*; *Sassoubre et al., 2016*). Therefore, we investigated the eDNA signal detection of four morphologically different NIS in the sampling region (*Woods et al., 2022*), ranging from leathery tunicates to soft algae tissue, to determine appropriate submersion times in relation to the targeted species and for the different materials tested.

Species-specific ddPCR assays were applied for all four NIS to ensure sensitive and quantifiable eDNA signal detections (*Wood et al., 2019*; *von Ammon et al., 2019*). Our results demonstrated that passive samplers could match or even surpass the eDNA yields obtained through seawater filtration (the conventional approach for collecting waterborne eDNA), confirming their efficiency and suitable application for surveillance purposes. While active filtration captures a finer snapshot of eDNA suspended in the water column (*Chen et al., 2024*), passive samplers accumulate signals over a certain deployment time (*Zhang et al., 2024*). The accumulation effect can lead to increased eDNA signals and

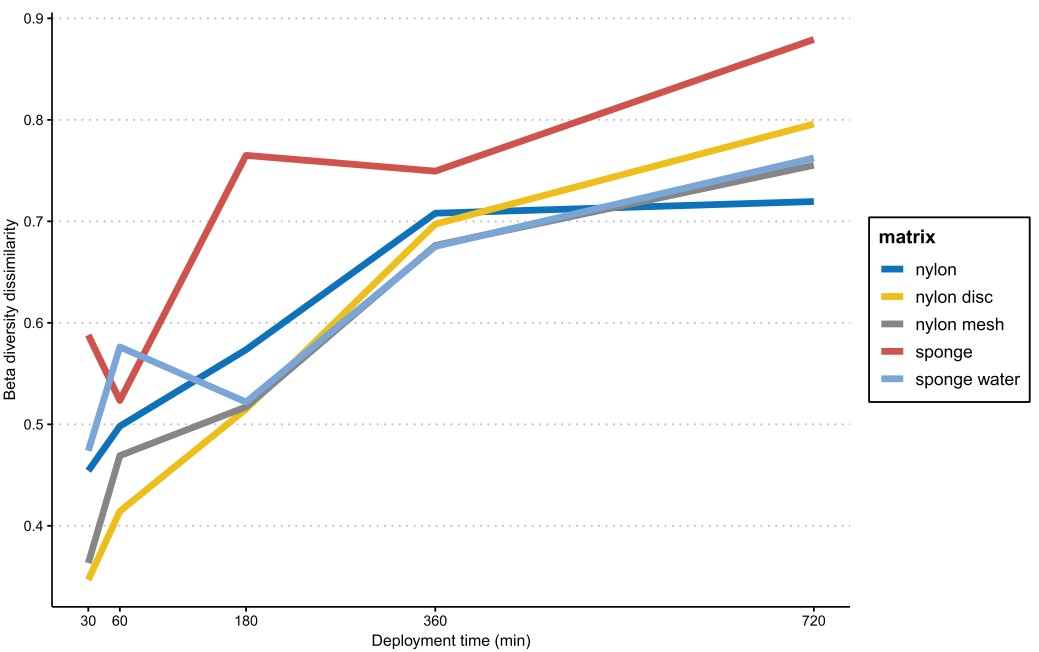

**Figure 6 Progression of beta-diversity dissimilarity.** Progression of beta-diversity dissimilarity of samples from 30 to 720 min compared to community composition at the beginning of the experiment (10 min).

recent studies have indicated significant correlations between increased eDNA yields and passive sampler deployment times, particularly with glass fiber and cellulose-based filters (*Chen et al., 2022*). In line with these findings, we observed significant trends in eDNA signal accumulation for the Mediterranean fanworm (*S. spallanzanii*) and the clubbed tunicate (*S. clava*), with the highest signals detected at the final sampling time. *Sabella spallanzanii* showed the strongest amplification signals across all tested assays, likely due to its high abundance at the sampling site and the species' reported higher eDNA shedding rates (*Wood et al., 2020*). The brittle body structure of the fanworm also results in more intensive release of eDNA particles into the water column in comparison to the leathery and compact tunicate, which reportedly releases significantly less detectable material (*Scriver et al., 2024a*). However, extreme outliers for all taxa may occur due to the stochastic nature of eDNA data (see *S. clava* in sponge water).

Surprisingly, the brown algae *U. pinnatifida* was not detected despite being known to occur in the study region. It is possible that individuals were not directly present near the sampling site, or that the lack of detection represents a false negative result. False negatives are a common problem with eDNA technologies, which can be associated with insufficient sampling, low DNA recovery, inhibition or insufficient shedding rates due to the organisms' morphologies (*Gold et al., 2021*). For example, *Waters et al. (2023)* showed how a validated assay failed to detect the green algae *Caulerpa prolifera* in the field. Similarly, *Kirtane, Atkinson & Sassoubre (2020)* found measurable total DNA yields on their passive samplers but the targeted freshwater mussel *Lampsilis* sp. could not be amplified. These

examples highlight the critical need for species-specific benchmarking of eDNA assays (*Waters et al., 2023*).

Here, the weak and scattered signals for the invasive bryozoan *B. neritina* did not indicate any accumulation of eDNA signal on passive samplers over time. Mesocosm experiments by *Scriver et al. (2024a)* confirmed reduced eDNA detection for this species, which is possibly related to low shedding rates, inhibitory effects from its calcareous body structure or comparatively low biomass (*Furlan & Gleeson, 2017*). To optimize eDNA detection efficiency across species, combining different sampling approaches and developing more sensitive molecular assays targeting mitochondrial or chloroplast markers could be beneficial (*Hunter et al., 2019*; *Krehenwinkel, Pomerantz & Prost, 2019*). However, it is essential to acknowledge the limitations of eDNA tools in detecting organisms with low shedding rates (*Ahyong & Wilkens, 2011*).

Despite the lower specificity of 18S rRNA metabarcoding compared to ddPCR assays, the data provided important insights into patterns of community-level detection by passive sampling. While the resolution of the 18S rRNA marker is not adequate for species assignation, it covers a broad range of taxa and has been effective in detecting NIS for marine biosecurity applications (*Borrell et al., 2018*), *e.g.*, in combination with COI (*von Ammon et al., 2023*; *Zaiko et al., 2023*). When assessing which of the targeted NIS could be detected in the metabarcoding data, the three taxa *S. spallanzanii*, *U. pinnatifida* and *B. neritina* were not detected. Previous studies have shown that *S. spallanzanii* is amplified using the COI marker and the whole group of bryozoan were not amplified by the 18S rRNA gene (*von Ammon et al., 2018b*). For algal taxa, rbcL and matK are more preferable marker choices (*Bartolo et al., 2020*). The genus *Styela* could sporadically be detected without differentiating between the species *clava* and *plicata*. However, the purpose of this study was not to compare the metabarcoding data with species-specific detections as previously done by *Wood et al. (2019)* and *von Ammon et al. (2019)*, but to understand the general community settling on the passive samplers over the deployment times.

Both material type and submersion times had a significant effect on community composition, which was also observed in a similar study on groundwater ecosystems (*van der Heyde et al., 2023*). Expectedly, taxonomic composition of the active water filtration samples was clearly dominated by planktonic families such as Calanoida and Tachidiidae (copepods), highlighting the importance of including water samples if the targeted organism has a swimming state. The observed shift between the first and last sampling time may be related to the vertical migration associated with the diurnal cycle of planktonic taxa (*Blanco-Bercial, 2020*). These fine scale temporal patterns can only be detected through an active sampling approach to provide more of a snapshot at the time of sampling.

For the passive samplers, sessile biofouling organisms such as ascidians and polychaetes were dominating and increased in abundance after 10 to 30 min, displaying an effect in accumulating biomass. Similar to the effect on settlement plates, passive sampler surfaces attracted sessile species (*e.g.*, barnacles, ascidians and polychaete larvae) that are known to attach within just a few seconds of submersion. These biofouling organisms became more

dominant over time, increasing in relative abundance while reducing overall biodiversity richness. Many of these sessile taxa are non-indigenous species and therefore of particular interest for biosecurity surveillance (*Tait & Inglis, 2014*). Despite the observed accumulation effect, species-specific detection signals (*e.g.*, the ddPCRs for the polychaete *S. spallanzanii* and the tunicate *S. clava*) did not seem affected, as longer deployment times equally showed an accumulation effect for the species-specific detection signals of the two biofouling organisms. However, several studies have recommended not to exceed 24 h of submergence time to prevent biofilm growth, which is associated with DNA degradation and signal loss through water movement (*van der Heyde et al., 2023*). Our experiment did not exceed 12 h and therefore did not show such effects, indicating instead an increase in species-specific detections with longer deployment times.

Therefore, we conclude that the choice between passive samplers and active filtration for eDNA surveillance depends on the study objectives, environmental conditions, logistics, and costs. In remote locations and with low budget, passive samplers offer a valid sampling strategy that can accumulate rare signals over time. Active filtration provides more precise data regarding sample volume and snapshot in time, however if benthic or sessile organisms are targeted, sampling from sediment or settlement plates should also be considered. Generally, for biosecurity and surveillance applications it is always advisable to complement metabarcoding-based detections with species-specific assays, and ideally validate with visual/physical detections to warrant active management actions (*Darling et al., 2020*).

## CONCLUSION

Significant variation in eDNA signal detection highlight the importance of considering both material selection and submersion time for passive eDNA samplers depending on the targeted organism. We demonstrated that passive samplers could achieve comparable or even higher eDNA yields than active filtration methods, suggesting their potential as efficient tools for NIS detection in marine environments.

Species-specific ddPCR assays provided sensitive and quantifiable eDNA signals for targeted NIS. However, our results underscore the importance of assay validation and the potential for false negatives, highlighting the need for rigorous testing and optimization of eDNA assays in the field.

Furthermore, 18S rRNA metabarcoding revealed that certain sessile organisms like ascidians and polychaetes started dominating communities detected on passive filters early on but did not interfere with the species-specific detection signals.

In summary, by considering material selection, submersion time, and assay validation, researchers can enhance the sensitivity and reliability of eDNA-based monitoring approaches, ultimately contributing to effective marine biosecurity and conservation efforts.

## ACKNOWLEDGEMENTS

We thank the captain (Jochen Zaeschmar) and Far Out Ocean Research Collective crew of the SV Manawanui for their participation in this research expedition and Marsden Cove

for letting us conducting our experiments. ChatGPT was used to streamline code for biostatistical analysis.

### Funding
This study was supported by New Zealand Ministry of Business, Innovation and Employment funding (CAWX1904—A toolbox to underpin and enable tomorrow's marine biosecurity system). The funders had no role in study design, data collection and analysis, decision to publish, or preparation of the manuscript.

### Grant Disclosures
The following grant information was disclosed by the authors:
New Zealand Ministry of Business.
Innovation and Employment funding: CAWX1904.

### Competing Interests
Xavier Pochon is an Academic and Section Editor for PeerJ. Anastasija Zaiko is co-founder and lead scientist of Sequench Ltd.

### Author Contributions
- Ulla von Ammon conceived and designed the experiments, performed the experiments, analyzed the data, prepared figures and/or tables, authored or reviewed drafts of the article, and approved the final draft.
- Gert-Jan Jeunen conceived and designed the experiments, performed the experiments, authored or reviewed drafts of the article, and approved the final draft.
- Olivier Laroche analyzed the data, prepared figures and/or tables, authored or reviewed drafts of the article, and approved the final draft.
- Xavier Pochon performed the experiments, authored or reviewed drafts of the article, and approved the final draft.
- Neil J. Gemmell performed the experiments, authored or reviewed drafts of the article, and approved the final draft.
- Jo-Ann L. Stanton conceived and designed the experiments, performed the experiments, authored or reviewed drafts of the article, and approved the final draft.
- Anastasija Zaiko conceived and designed the experiments, performed the experiments, authored or reviewed drafts of the article, and approved the final draft.

### Field Study Permissions
The following information was supplied relating to field study approvals (*i.e.*, approving body and any reference numbers):

Field experiments were approved by the Cawthron Institute. Cawthron holds a Special Permit with the Ministry for Primary Industries (SP822-2) that allows the taking of fish, aquatic life and seaweed for the purposes of education and investigative research.

## Data Availability

Raw sequence reads were deposited in the NCBI short read archive under the accession number PRJNA1154297.

## Supplemental Information

Supplemental information for this article can be found online at http://dx.doi.org/10.7717/peerj.19043#supplemental-information.

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
