# Peer review of "Investigating passive eDNA samplers and submergence times for marine surveillance"

_PeerJ, doi:10.7717/peerj.19043_

## Round 0.1 · original submission · Major Revisions

Dear Authors, three reviewers have submitted their comments and suggestions concerning your paper. All of them have found the paper well written, with a clear and well-described aim, and a good coverage of all relevant literature. They gave you several tips and specific comments that will help to improve the manuscript. My decision is to further revise the manuscript and resubmit it.

Reviewer 1 ·

Basic reporting

The article is well written and clearly presents the authors' main findings which achieve their objectives. The literature is sufficiently cited and the figures are of high quality.

I did notice a few small grammatical errors that should be corrected:

Line 66: Change “traditional” for conventional, since traditional can imply cultural tradition.

Line 152: This sentence appears to be missing a verb.

Line 249: There is an extra parentheses here.

Line 295: "After discarding negative controls." is not a complete sentence.

Further, I have some comments regarding the figures:

Figure 1: The map in Figure 1 does not include longitude. An image of the bottle carrier deployment device would be helpful for those who are not experienced in using passive filtration methods.

Figure 2: Is it possible to plot the y axis on a squareroot scale instead of plotting the transformed numbers? That way the actual values before transformation could be visualized. If left as is, the y axis should more accurately read √Copies/μl

Figure 3: The trend lines do not appear to show linear regressions. This should be corrected and also clarified in the methods.

Figure 5: Species names should be italicized in the legend.

Experimental design

The experimental design is appropriate and comprehensive to answer the authors' research question; however, some key points are missing.

First, the limits of detection for the digital droplet PCR assays are not reported. Admittedly, I am more familiar with reporting limits of quantification and limits of detection with quantitative PCR, but it appears to be standard practice to report limits of detection for ddPCR as well.

Second, it is not clear whether or not the primers for U. pinnatifida were designed in this study or designed previously. How were the primers designed and how was this assay tested for specificity? A table could be included with all of the primer sequences and their citations.

Third, the authors have taken necessary negative control samples at each step in sampling processing, including field blanks and PCR negative controls. They should be commended for these efforts. However, the results of these negative controls are not sufficiently reported. The authors do not report whether any negative controls amplify with the ddPCR assays. How many copies were were measured if they did? Table S1 shows that filtration blanks, extraction blanks, and PCR negative controls all produced a high number of sequencing reads in the 18S data, on par with those from the field samples. While the authors report the ASVs that were detected in the negative controls (Table S2), they do not report their abundance in the negative control samples, only their abundance in the field samples. It is not clear how the presence of certain taxa (e.g. Styela plicata) in the negative controls that were also abundant in the field samples was accounted for. How did microdecon remove contaminant sequences? Was this used to correct for cross contamination?

Finally, the statistical tests performed require some clarification. For example, Figure 3 is supposed to show the results of linear regressions, but the trend lines fit to the data are non linear. For these linear regressions, should R^2 values be reported instead of R values? In line 244: What statistical tests were performed to determine if there were changes in ASV richness with deployment duration?

Validity of the findings

There seems to be evidence of prevalent cross contamination among negative control samples in the 18S data that is not sufficiently addressed. The results of the ddPCR negative control samples are not discussed, and are the ddPCR data are not included as a supplement or through a DOI link. The code to analyze the data is also not included.

The results are very well presented; however, is difficult to assess their validity unless these points are addressed.

Reviewer 2 ·

Basic reporting

Overall the necessary information is adequately described; "Introduction" and "Discussion" sections well described the purpose of the study and the previous findings on passive sampling with references to the literatures, although ”Material and Methods” and ”Results” sections need some minor corrections to be made, as suggested in "2. Experimental design".
For "Discussion", there are some paragraphs where they are not clear which results were being addressed. For example, it is difficult to understand on which results L363-367 is based. It should be stated in the main text which Figs or Tables they correspond to.

L421-423 and L428-430
The results are interesting because they clearly show the difference in trend between active water filtration sampling and passive sampling. It would be useful information to consider when using these different methods in the future. What tends to happen in passive samplers could be explained in more detail with the results. For example, the interpretation of the decrease in AVS richness while absolute/relative abundance of specific species (Fig. 2/Fig. 5) increased should be explained.

L438-440
Fig. 6 shows that beta-diversity has increased over time from 10 min, suggesting species composition has changed over time, and Fig. 4 shows that ASV richness decreased with time. It looks a decrease in biodiversity as deployment time increases.

Experimental design

The research objectives are clear and the experimental methods are in line with them, but the following minor corrections should be made;

L175
Which reference was used for the primer and probe sequences for Undaria pinnatifida?

L168-184
In ddPCR for Sabella spallanzanii, it states that the COI is the target region, but the target regions of the other three primer (probe) sets should also be listed.

L236 NIS and L243 ASV
They appear here for the first time and should be spelt out in full rather than as abbreviations only.

L331~ and Fig. 6.
Is the beta-diversity of filtered water samples more variable between start-end?

Fig. 2
The vertical axis of the bar chart for Styela clava does not appear to match the values given in the main text. For example, the signal intensity for nylon mesh (1h) does not appear to be 0.059 copies/µl. Is this due to a difference between mean and median values? The discrepancy between the figure and the text causes confusion for readers.

Fig. 3
What horizontal dashed lines mean should also be stated here.

Fig. 4.
What horizontal dashed lines mean should also be stated here. In Material and Methods and Fig 5 show where the NIS was obtained, but it should be explained also in Fig. 4 and the corresponding main text.

Legends of Supplementary Figs should be added.

Validity of the findings

While surveys using waterborne eDNA in fish are a well-established method, this paper is significant in that validation data using passive sampling, which has recently been gaining attention for invertebrates and algae with different morphologies and biomasses, is useful for the control of invasive species which are relatively difficult to detect. As mentioned in “1. Basic reporting”, the identification of trends in passive sampling should provide recommendations on how it can be applied to conservation and invasive species control in the future, and could be discussed further.

·

Basic reporting

Very clear and concise writing with good coverage of all relevant literature. Raw data is shard on SRA. Hypotheses could be better defined with rationalizations e.g was the reduction in ASV richness over time expected?

Experimental design

Very good experimental design to address the research questions.

Validity of the findings

The contrary results between the ddPCR and metabarcoding could be explored further that would make this article more interesting. It would also be interesting to know what aspects of the study the authors would encourage replication for, are both materials and submergence times equally important as independent variables in future studies? Should future studies use both ddPCR/qPCR and metabarcoding for their assessments ? What might be the risks of only using one metric as seen in this study.

Additional comments

I have reviewed the manuscript titled “Investigating passive eDNA samplers and submergence times for marine surveillance.” The manuscript builds on valid research questions raised by previous studies: How is the efficacy of passive samplers impacted by submergence times, and do the sampling materials influence these trends? These questions were investigated via empirical testing with ddPCR for the detection of four invasive species, as well as 18S rRNA metabarcoding. This study is clear and well-written. I have only suggested minor edits in the text.

However, the data from ddPCR and metabarcoding seem to show opposite trends. This is quite interesting and should be highlighted more clearly in the text and discussed in greater detail to reduce the potential for confirmation bias as readers evaluate this highly interesting and relevant study. I believe that with these concerns addressed, this study is fit for publication in PeerJ.

Specific Comments:
Line 85: The sentence cites two studies, but a third study’s results are elaborated on in the following sentence to back up the claim.
Line 127: This part needs to be explained better: “and additionally the water squeezed from the Whirl-PakÆ Speci-Sponges.”
Line 236: NIS needs to be defined.
Line 242: Was the sample data rarefied? It says that they were rarefied in line 247, but no threshold is mentioned.
Line 322: Did NIS detection increase over time with passive sampling? This seems confusing, as line 303 mentions that NIS decreased with time. Perhaps this needs more clarification.
Line 397: Does this refer to eDNA yields or copy number of targeted NIS?
Methodological Concerns:
ddPCR methods/results: Assay kinetics and LOD/LOQ must be reported based on standardized guidelines. See:
Klymus, Katy E., et al. "Reporting the limits of detection and quantification for environmental DNA assays." Environmental DNA 2.3 (2020): 271-282.
Figures:
Figure 3: The regressions are not linear. Is the relationship between eDNA accumulation and time expected to be non-linear/exponential? What is the mechanistic rationale for choosing the given regression method (GLM with exponential link function) for this application?
Figure 4B: Do these scatterplots also include the NIS detected by the ddPCR assays? Were they co-detected with both analytical methods but showed different trends with submergence time?
The ASV richness seems to decline with submergence time, while the ddPCR detection of invasive species eDNA seems to increase in most cases (Figures 3, 4). Authors should discuss this in greater detail. Could this all be attributed to degradation and biofouling? It would also be interesting to see the ASV richness of the active filtration samples alongside these plots.
Figure 5B: What are the white spaces in Figure 5B? Relative abundance plots should not have white spaces. Is it all B. leachii? Are there non-invasive taxa represented here?
Figure 6: Interesting! The beta community composition at longer submergence time points is more different from the initial time point. This might be due to reduced ASV richness over time, as shown in Figure 4. However, the increasing beta dissimilarity, as shown in Figure 6, could be easily misinterpreted as a function of increased diversity captured over submergence time due to intuitive hypotheses and signals from Figure 4. I would encourage the authors to use a different visualization to convey this information or add additional details to avoid misinterpretation.

---

## Round 0.2 · Minor Revisions

Dear Authors, the reviewers read the manuscript and both of them agree that you have addresses all their comments. There are very few typos to fix before publication. Lines 106-107: a reference will be nice to add to confimr the statement concerning the abundance of these species in the region; Line 240: delete end point; Line 257: a parenthesis is missing; LIne 448: Calanoida and Tachidiidae not in italic; line 462: and the tunicate NOT in italic; line 467: delete ".

Reviewer 1 ·

Basic reporting

My previous comments regarding the text and figures have all been addressed, and the manuscript is in great shape. I commend the authors for the quality of the initial submission and for their attention to detail when responding to reviewer comments.

I did notice a few small grammatical errors that should be fixed.

Line 259: There is an extra period at the beginning of this line.

Line 410: Should say "...freely deployed as a barrier AND may inadvertently..."?

Line 483: Extra space between "data" and "provided"

Lines 520-522: Includes an extra " as well as an extra period.

Experimental design

The authors have sufficiently addressed my comments regarding ddPCR primer design and how contamination was dealt with in the metabarcoding data.

Validity of the findings

The authors have sufficiently described the steps taken to mitigate contamination, and have now provided the raw ddPCR data and the code necessary to analyze it.

Reviewer 2 ·

Basic reporting

The suggestions from the reviewers were corrected properly and the newly added sentences and Supplementary table made this manuscript clearer. I think the revised manuscript is almost acceptable for publication, but I would like to add the following minor suggestion;

As for my previous comment "For Discussion, there are some paragraphs where they are not clear which results were being addressed.", L449-450, L470-471, L498-499, L505-506 and L513-517 could also state which Fig/Table correspond to the discussions, so that the reader is not confused as to which results to refer to.

Experimental design

no comment

Validity of the findings

no comment

Additional comments

no comment

---

## Round 0.3 · accepted · Accept

Dear Authors, I am happy to inform you that your paper is accepted for publication in PeerJ